# Identification and Functional Analysis of *foxo* Genes in Chinese Tongue Sole (*Cynoglossus semilaevis*)

**DOI:** 10.3390/ijms24087625

**Published:** 2023-04-21

**Authors:** Tingting Zhang, Mengqian Zhang, Yuxuan Sun, Lu Li, Peng Cheng, Xihong Li, Na Wang, Songlin Chen, Wenteng Xu

**Affiliations:** 1Yellow Sea Fisheries Research Institute, Chinese Academy of Fishery Sciences (CAFS), Laboratory for Marine Fisheries Science and Food Production Processes, Pilot National Laboratory for Marine Science and Technology, Qingdao 266071, China; 2School of Fisheries and Life Science, Shanghai Ocean University, Shanghai 201306, China

**Keywords:** *Cynoglossus semilaevis*, *foxo*, promoter activity, siRNA knockdown

## Abstract

The Chinese tongue sole (*Cynoglossus semilaevis*) is a traditional, precious fish in China. Due to the large growth difference between males and females, the investigation of their sex determination and differentiation mechanisms receives a great deal of attention. Forkhead Box O (*FoxO*) plays versatile roles in the regulation of sex differentiation and reproduction. Our recent transcriptomic analysis has shown that *foxo* genes may participate in the male differentiation and spermatogenesis of Chinese tongue sole. In this study, six *Csfoxo* members (*Csfoxo1a*, *Csfoxo3a*, *Csfoxo3b*, *Csfoxo4*, *Csfoxo6-like*, and *Csfoxo1a-like*) were identified. Phylogenetic analysis indicated that these six members were clustered into four groups corresponding to their denomination. The expression patterns of the gonads at different developmental stages were further analyzed. All members showed high levels of expression in the early stages (before 6 months post-hatching), and this expression was male-biased. In addition, promoter analysis found that the addition of C/EBPα and c-Jun transcription factors enhanced the transcriptional activities of *Csfoxo1a*, *Csfoxo3a*, *Csfoxo3b*, and *Csfoxo4.* The siRNA-mediated knockdown of the *Csfoxo1a*, *Csfoxo3a*, and *Csfoxo3b* genes in the testicular cell line of Chinese tongue sole affected the expression of genes related to sex differentiation and spermatogenesis. These results have broadened the understanding of *foxo*’s function and provide valuable data for studying the male differentiation of tongue sole.

## 1. Introduction

In mammals, the forkhead box (*fox*) family has been reported to consist of over 40 members, including *foxa-q*. Forkhead box O (*foxo*) is a subgroup of the forkhead box transcription factor family [1]. It is the evolutionarily conserved transcription factor that was first identified in Drosophila melanogaster in 1989 [2,3]. *foxo* plays a major role in various cellular processes, such as cell differentiation and growth, apoptosis, proliferation, oxidative stress, and DNA damage repair [4,5]. *foxo* is regulated by two signalling pathways. The first is the classical insulin signaling pathway, which is regulated by phosphatidylinositol-3-kinase (PI3K) and protein kinase B (*protein kinase B, AKT*) in the presence of growth factors. The second pathway is c-Jun N-terminal kinase (*c-Jun N-terminal kinase, JNK*), which acts during oxidative stress [4,6]. Four subtypes of *foxo* have been identified in mammals: *foxo1, foxo3, foxo4,* and *foxo6*, while in bony fish, seven subtypes, *foxo1a*, *foxo1b*, *foxo3a*, *foxo3b*, *foxo4*, *foxo6a*, and *foxo6b,* are well known [7,8]. In zebrafish, *foxo2* and *foxo5* have also been reported [5]. However, *foxo2* is a paralogue of *foxo3*, while *foxo5* is specifically in zebrafish and is also called *foxo3b*.

Foxo family members are characterized by the presence of a 100-amino acid conserved DNA binding domain [9]. This domain exists in several different transcription factors and determines cell fate during the early stage of embryogenesis [10]. The most frequently reported members are *foxo1* and *foxo3*. According to a previous transcriptome analysis [11], *foxo1* is believed to play a role in spermatogenesis and spermatogonia differentiation in mammals [12]; for example, *foxo1* regulates the maintenance of male spermatogonia stem cells in mice (*Mus musculus*) [13]. In fish, *foxo1a* is reported to function in the insulin-related pathway and growth of rainbow trout (*Oncorhynchus mykiss*), grass carp (*Ctenopharyngodon idella*) [14,15], and spotted seabass (*Lateolabrax maculatus*) [16]. *foxo3* is a key effector of cell death in mammals [17]. *foxo3* can inhibit follicle growth initiation and control reproductive potential. In the orange-spotted grouper (*Epinephelus coioides*), *foxo3a* and *foxo3b* proteins exist in ovarian germ cells and follicular cells, which may be involved in follicular formation [18]. In zebrafish, *foxo4*, *foxo6a*, and *foxo6b* are reported to function in the developing brain and under the regulation of PI3-kinase signalling [19].

The Chinese tongue sole exhibits obvious sexual growth dimorphism, wherein females can grow two to four times larger than males, so the study of sex differentiation has great potential for application in aquaculture. Many *fox* members have been reported to function in sex determination and differentiation, such as the classical *foxl2* and *foxl3* [20]. Due to their important role in sex differentiation and growth, *foxo* genes were selected for an in-depth analysis of the Chinese tongue sole. In this study, the conserved domain, phylogenetic tree, and conserved motifs of six *Csfoxo* members were analyzed. Their expression patterns in gonads at different developmental stages were also studied. The transcriptional regulation of *Csfoxo1a*, *Csfoxo3a*, *Csfoxo3b*, and *Csfoxo4* was then analyzed. Finally, the siRNA-mediated knockdown of *Csfoxo1a*, *Csfoxo3a*, and *Csfoxo3b* and its effect were investigated. These results may improve our understanding of the role of *foxo* genes in the Chinese tongue sole.

## 2. Results

### 2.1. Identification and Analysis of Csfoxo Genes

Six *Csfoxo* genes were identified in the tongue sole genome. Their gene IDs, chromosome localizations, amino acid numbers, molecular weights (MWs), and isoelectric points (pI) were analyzed (Table 1 and Appendix A). The ORF sequences of these genes varied from 1872 to 2178 bp in length and encoded 623-725 amino acids. The predicted MWs were 66.52–78.07 kDa with pI values ranging from 4.79–6.78. As shown in Figure 1, all the proteins share a forkhead domain and contain low-complexity regions (LCRS) with little variation among different members.

### 2.2. Phylogenetic Tree Analysis

We performed a multiple alignment procedure, which showed the similarity of the six Foxo proteins is 53.1%. To study the phylogenetic relationships, an NJ tree was constructed based on the amino acid sequences of nine fish and two mammals. As shown in Figure 2, 53 Foxo proteins were divided into four clusters (Foxo1, Foxo3, Foxo4, and Foxo6), indicating the evolutionary relationship between *foxo* members in vertebrates. Only one copy was found in mammals, whereas multiple copies of Foxo1, Foxo3, and Foxo6 were identified in fish. It is worth noting that inside each of the clusters, the Foxo proteins in fish and mammals formed different subclusters.

### 2.3. Structure Analysis of Foxo

As shown in Figure 3, a total of 12 motifs were detected, and the signature sequences of the 12 motifs are shown in Appendix A. Motifs 1 and 2 are highly conserved FH domains, and motif 9 is only found in Foxo3 and Foxo4 of bony fish, consisting of 50 amino acids. In fish, Foxo3 contains all motifs, while Foxo4 lacks motif 10, and Foxo1 and Foxo6 lack motif 9. In mammals, the Foxo motifs differ among humans, mice, and rats (Appendix A). In general, mammalian Foxo1 and Foxo3 lack motif 11, Foxo4 lacks motif 8, and Foxo6 lacks motifs 8, 10, and 12.

### 2.4. Analysis of Csfoxo Expression Patterns at Different Developmental Stages

As shown in Figure 4, the expression patterns of six *Csfoxo* members in the gonads at different developmental stages were analyzed. All *Csfoxo* members showed higher expression before 6 mph, and this expression was male-biased. However, the peak levels of different members occurred at different stages, e.g., *foxo6-like* at 40 dpf, *foxo3a*, *foxo3b* and *foxo4* at 60 dpf, and *foxo1a* and *foxo1a-like* at 90 dpf. These data suggest their involvement in male differentiation and testicular development.

### 2.5. Promoter Activity Analysis

As shown in Figure 5, the promoter regions of the *Csfoxo1a*, *Csfoxo3a*, *Csfoxo3b*, and *Csfoxo4* genes were cloned to explore promoter activity. The promoter activity of pGL3-*Csfoxo1a*, *3a*, *3b*, and *4* was significantly higher than that of pGL3-basic, indicating that the *Csfoxo* promoter had a positive effect on *Csfoxo* gene expression. The activity of pGL3-*Csfoxo1*, *3a*, *3b*, and *4* was significantly promoted by adding C/EBPα and c-Jun transcription factors, suggesting that C/EBPα and c-Jun could positively regulate *foxo* genes. This activity was closely related to the specific site. When the C/EBPα and c-Jun binding sites were mutated (muCsfoxo1a-C/EBPα, muCsfoxo3a-C/EBPα, muCsfoxo3b-C/EBPα, muCsfoxo4-C/EBPα, muCsfoxo1a-c-Jun, mufoxo3a-c-Jun, mufoxo3b-c-Jun, and mufoxo4-c-Jun), the levels of activity were significantly decreased.

### 2.6. Csfoxo Knockdown and the Effect on Other Genes

For *Csfoxo1a*, *3a*, and *3b*, three siRNAs were designed for each gene. The knockdown effect was examined, and the siRNAs with the most obvious effect (siRNA1 for *Csfoxo1a*, siRNA1 for *Csfoxo3a*, and siRNA3 for *Csfoxo3b*) were selected (Figure 6A,C,E). After the knockdown of *Csfoxo1a*, sex-related genes, including *sox9a* and *wt1a*, and the growth-related gene insulin-like growth factor *igf1* were downregulated (Figure 6B). Knockdown of *Csfoxo3a* induced the downregulation of *tesk1* (a spermatogenesis-related gene) and *wt1a* (Figure 6D). After *Csfoxo3b* knockdown, the genes *igf1, neurl3* (a spermatogenesis-related gene)*, sox9a*, and *wt1a* were downregulated (Figure 6F).

## 3. Discussion

*foxo* is involved in a wide range of biological processes, including cell differentiation, metabolism, tumor inhibition, cell cycle arrest, protection from stress, and cell death [21,22,23]. However, the functionality of *foxo* genes has not been systematically studied in the Chinese tongue sole. In this study, we first focused our attention on *Csfoxo* based on our previous transcriptomic analysis [11]. We identified six *foxo* members in the Chinese tongue sole: *foxo1a*, *1a-like*, *3a*, *3b*, *4*, and *6-like*. In comparison to seven members present in bony fish, one copy of *foxo6* seems to be missing in tongue sole [6]. The Csfoxo members share the conserved FH binding domains of “winged helix” or “forkhead box”, which can bind to B-DNA as monomers [6]. A low-complexity region (LCR) is found in all Csfoxo members; it may participate in flexible binding, and its position in the protein may determine its binding properties and specific function [24]. Structural and phylogenetic tree analysis indicate that Foxo is highly conserved in vertebrates. It is interesting that all six members showed high expression before 6 mph and were male-biased, suggesting their role in male differentiation and testicular development. However, their peak levels appeared at different stages, e.g., *foxo6-like* appeared at 40 dpf, *foxo3b* and *foxo4* at 60 dpf, *foxo1a* and *foxo1a-like* at 90 dpf, and *fox3a* at 6 mpf. Whether they play a sequential role in male differentiation and testicular development requires further investigation.

CCAAT/enhancer-binding protein α (C/EBPα) is an evolutionarily conserved transcription factor in vertebrates that plays a role in cell growth and differentiation [25]. In Chinese tongue sole, C/EBPα was reported to be involved in early gonadal differentiation and to act as a negative regulator of the male-determining gene *dmrt1* [26]. c-Jun is a transcriptional activator that plays an important role in cell proliferation and differentiation [27]. Based on our data, C/EBPα and c-Jun enhanced the promoter activity of all *foxo* members. However, the interaction of *foxo* with other sex-related genes (especially *dmrt1*) requires further investigation.

To investigate the effects of *foxo* genes, several growth- and sex-related genes were selected, including *igf1, sox9a*, *wtla*, *tesk1*, and *neurl3*. As a well-known growth-related gene, *igf* is also reported to play a role in sex differentiation. In tilapia, *igf1* is distributed in the gonads at early stages [28,29], and it may be involved in the regulation of growth and differentiation [30,31]. In mammals, *sox9* plays a role in cell differentiation and male differentiation [32]. In fish species, such as rainbow trout and zebrafish, *sox9a* showed higher expression in the testis [33,34]. Wilms tumor gene, *wt1a,* has multiple pivotal roles in gonads. In adult medaka, the *wt1a* gene is highly expressed in Sertoli cells of the testis and is required for PGC maintenance [35]. In tongue sole, the alternative splicing of *wt1a* has been suggested to play a role in gonadal differentiation [36]. *Tesk1* plays a role in spermatogenesis in mice, especially during early spermatogenesis [37]. In Chinese tongue sole, *tesk1* and *neurl3* are closely related to spermatogenesis [38]. It is worth noting that the knockdown of *foxo1a, foxo3a*, and *foxo3b* led to the downregulation of all the abovementioned genes, suggesting the positive regulation of *foxo* members in male differentiation, testis development, and spermatogenesis. In future studies, in vivo trials are still required for the in-depth investigation of the mechanisms of *foxo* genes in these processes.

## 4. Materials and Methods

### 4.1. Identification of Csfoxo Members and Sequence Analysis

The hidden Markov model (HMM) map of *foxo* TAD (PF16676) was downloaded from the Pfam protein family database (http://pfam.xfam.org/ (accessed on 4 May 2022)). Subsequently, Foxo genes from nine teleost fish, namely, tongue sole, medaka (*Oryzias latipes*), turbot (*Scophthalmus maximus*), zebrafish (*Danio rerio*), spotted gar (*Lepisosteus oculatus*), Nile tilapia (*Oreochromis niloticus*), fugu (*Takifugu rubripes*), pufferfish (*Perca fluviatilis*), and Atlantic halibut (*Hippoglossus hippoglossus*), and two mammals, mice and humans (*Homo sapiens*), were used for evolutionary analysis. The protein sequences were acquired from The National Center of Biotechnology Information (NCBI) (https://www.ncbi.nlm.nih.gov/ (accessed on 6 May2022)) database. The *Csfoxo* ProtParam tool (https://web.expasy.org/protparam/ (https://www.ncbi.nlm.nih.gov/ (accessed on 10 May 2022)) was used to analyze the chemical and physical properties of *Csfoxo*, including its number of amino acids, molecular weight (MW), and theoretical isoelectric point (pI).

### 4.2. Phylogenetic Tree and Structural Characterization

A phylogenetic tree was constructed based on the protein sequences of nine teleost fish and two mammals using MEGA v7.0. The adjacency (NJ) method and 1000 bootstrap replications were used. All protein sequences were downloaded from the NCBI database, and the information is shown in Appendix A. The conserved motifs were analyzed using MEME (5.4.1) (http://meme-suite.org/tools/meme
https://www.ncbi.nlm.nih.gov/ (accessed on 20 May 2022)) [39]. The number of motifs identified was 12, and other parameters were set to default values to obtain conservative motifs.

### 4.3. Experimental Animals and Ethics Approval

The Chinese tongue sole used in this experiment were from the breeding stock of Chinese tongue sole of Weizhuo aquatic company (Tangshan, China). MS-222 (20 mg/L solubilized in seawater, treated for 5 min) was used for anesthesia to minimize suffering before the gonads were dissected. Different developmental stages were selected, including 40 days post-hatching (dpf), 60 dpf, 90 dpf, 6 months post-hatching (mpf), 1.5 years post-hatching (ypf), and 3 ypf. For each stage, three male and three female individuals were examined. The gonads were quickly removed, placed into RNA preservation solution, and stored at −80 °C until RNA extraction was performed. The study was conducted under the inspection of the committee at the Yellow Sea Fisheries Research Institute (Approval number, YSFRI-2022035).

### 4.4. Gene Expression Pattern Analysis

TRIzol (Invitrogen, Carlsbad, CA, USA) was used to extract RNA from gonads at different developmental stages. The quality and quantity of RNA were determined using a P100 Series spectrophotometer (Pultton, San Jose, CA, USA). cDNA was synthesized using the Prime Script TM RT Reagent Kit with gDNA Eraser (Takara, Tokyo, Japan). Specific primers for *Csfoxo1a*, *Csfoxo3a*, *Csfoxo3b*, *Csfoxo4*, *Csfoxo6-like*, and *Csfoxo1a-like* were designed to analyze gonad expression in different developmental stages (Appendix A). A total reaction volume of 10 μL was used, including forward and reverse primers (0.2 μL each), 1 μL of cDNA, 5 μL of THUNDERBIRD^®^ Next SYBR^®^ qPCR Mix (TaKaRa, Tokyo, Japan), and 3.6 μL of ddH_2_O. qPCR was performed in a 7500 rapid real-time quantitative PCR system (Applied Biosystems, Foster City, CA, USA). The PCR program was set as follows: 95 °C dissociation for 30 s and 40 cycles of 95 °C for 5 s, and 60 °C dissociation for 30 s. β-a ctin was used as an internal reference. The relative transcription level of genes was calculated using the 2^−ΔΔct^ method [40]. The data were analyzed via one-way ANOVA and then using multiple comparisons in SPSS 25.0 (IBM Corp., Armonk, NY, USA), where differences were considered significant at *p* < 0.05.

### 4.5. Promoter Activity Analysis

Promoter plasmids (1500–3000 bp upstream of ATG) of *Csfoxo1a*, *Csfoxo3a*, *Csfoxo3b*, and *Csfoxo4* were constructed to detect dual-luciferase activity. The primers used are listed in Appendix A. The fragments were inserted into the pGL3-basic vector (Promega, Madison, WI, USA) using the TOROIVD^®^ One Step Fusion Cloning MIX Seamless cloning kit (Toroivd, Shanghai, China) to obtain pGL3-C-foxo1a, pGL3-C-foxo3a, pGL3-C-foxo3b, and pGL3-C-foxo4. Transcription factors of the C/EBPα gene and c-Jun amino terminal kinase gene were predicted to regulate foxo genes in the promoter region using Promo usage (http://alggen.lsi.upc.es/cgi-bin/promo_v3/promo/promoinit.cgi?dirDB=TF_8.3 (accessed on 30 May 2022)). According to the binding sites of transcription factors (C/EBPα and c-Jun binding sites), mutations in *foxo* were generated using the rapid site-directed mutagenesis kit (TIANGEN, Beijing, China). After successful mutation was confirmed via sequencing, transfection and double luciferase assays were performed.

Human embryonic kidney (HEK) 293T cells were maintained in DME/F-12 containing 10% foetal bovine serum (FBS, Gibco, New York, NY, USA) and 1% bFGF (Invitrogen, Carlsbad, CA, USA) in 5% CO_2_ at 37 °C. pGL3-foxo (pGL3-foxo1a, pGL3-foxo3a, pGL3-foxo3b, and pGL3-foxo4), pGL3-foxo+pcDNA3.1-C/EBPα, pGL3-foxo+pcDNA3.1-c-Jun, pGL3-control (positive control), and pGL3-basic (negative control) were transfected into 293T cells (800 ng plasmid/well). Lipo8000TM transfection reagent (Beyotime, Shanghai, China) was used, while the pRL-TK plasmid was transfected as an internal reference (40 ng/well), with three replicates per sample. At 48 h after transfection, 100 μL cell lysates from each well were collected using a double luciferase reporter assay kit (Beyotime, Shanghai, China) and analyzed with Origin 2021.

### 4.6. siRNA-Mediated Knockdown of Csfoxo in Testicular Cells

The siRNAs of *Csfoxo1a, Csfoxo3a*, and *Csfoxo3b* were synthesized by Sangon (Shanghai, China). Three siRNAs targeting different sites were designed for each gene, and the siRNA with the best knockdown effect was selected for further study. siRNA interference transfection was performed in a tongue sole testicular cell line (predominantly Sertoli cells) from our laboratory using a CP Reagent transfection kit (Ribobio, Guangzhou, China). According to the previously established experimental protocol, the *Csfoxo1a, Csfoxo3a, and Csfoxo3b* siRNA transfection and negative control (NC) transfection were performed in triplicate [41]. The cell status and fluorescence of cy3-transfected cells were examined 48 h after transfection. qPCR was used to detect the knockdown effect and the expression profiles of sex-related genes, such as *sox9a* and *tesk1*.

## Figures and Tables

**Figure 1 ijms-24-07625-f001:**
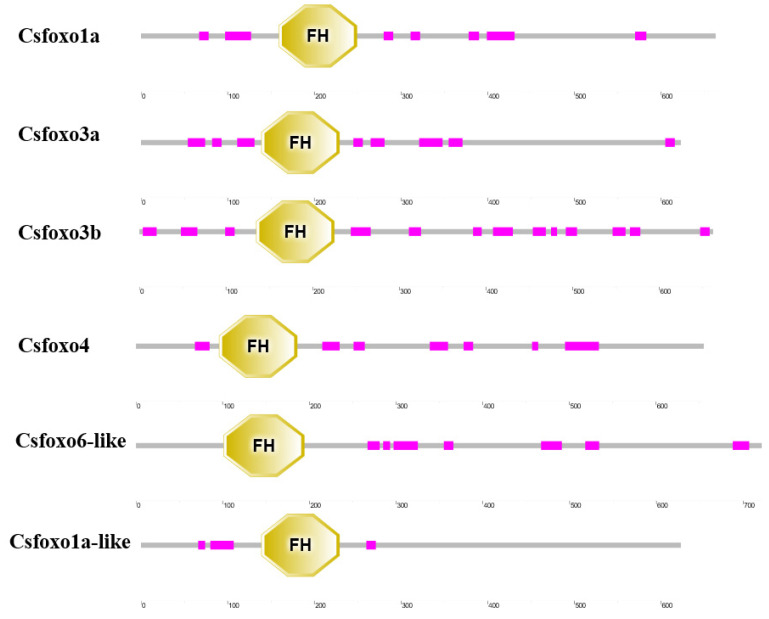
The conserved domain of the *Csfoxo* protein. Horizontal grey bars represent amino acid sequences with no predictive functional domains, while colored boxes represent regions with reliable predictive domains. Forkhead binding domain (yellow); Low-complexity region (pink).

**Figure 2 ijms-24-07625-f002:**
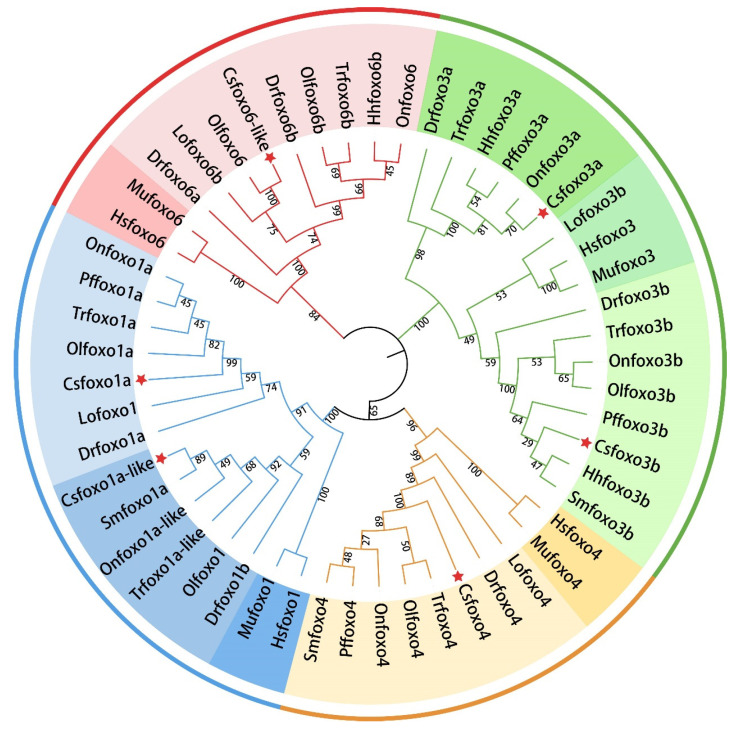
Phylogenetic trees of foxo from nine fish, Chinese tongue sole (Csfoxo), medaka (Olfoxo), turbot (Smfoxo), zebrafish (Drfoxo), spotted gar (Lofoxo), Nile tilapia (Onfoxo), fugu (Trfoxo), pufferfish (Pffoxo), and Atlantic halibut (Hhfoxo), and two mammals, mice (Mmfoxo) and humans (Hsfoxo). Different colors in the circle represent different *fox* members (blue for foxo1, red for foxo6, green for foxo3, orange for foxo4), and light color for fish and dark color for mammal. Red pentagrams indicate fox members of Chinese tongue sole.

**Figure 3 ijms-24-07625-f003:**
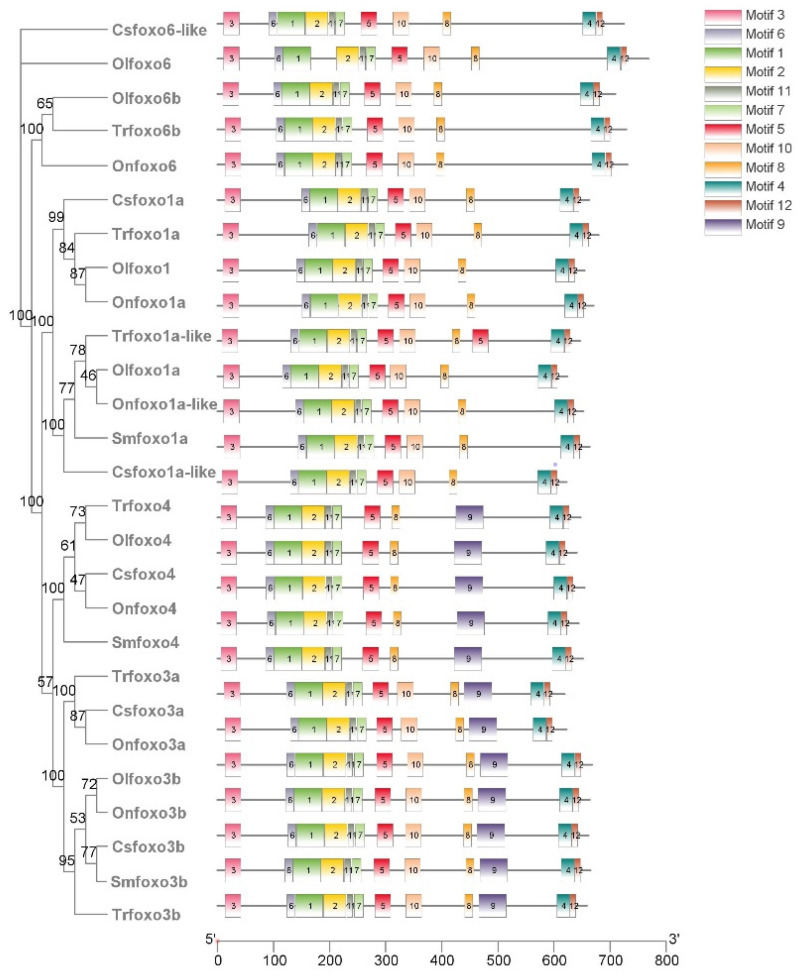
Structure and motif analysis of Foxo proteins in five fish species: Chinese tongue sole (Csfoxo), medaka (Olfoxo), turbot (Smfoxo), Nile tilapia (Onfoxo), and fugu (Trfoxo).

**Figure 4 ijms-24-07625-f004:**
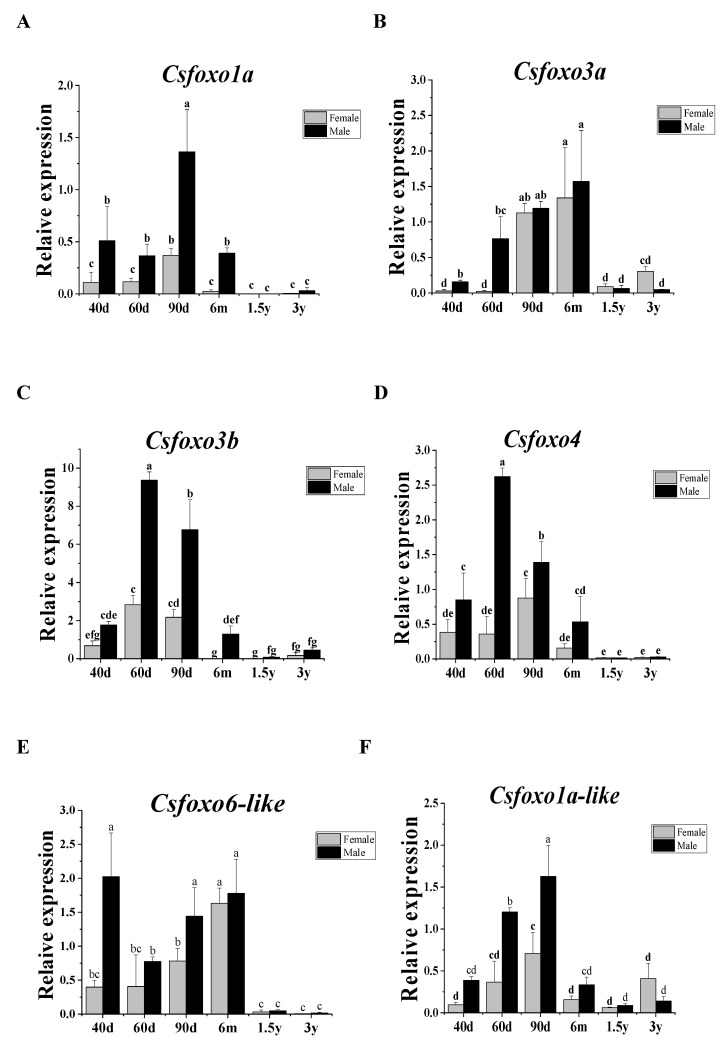
Expression patterns of *Csfoxo* in gonads at different developmental stages. (**A**) Expression patterns of *Csfoxo1a* at different developmental stages; (**B**) expression patterns of *Csfoxo3a* at different developmental stages; (**C**) expression patterns of *Csfoxo3b* at different developmental stages; (**D**) expression patterns of *Csfoxo4* at different developmental stages; (**E**) expression patterns of *Csfoxo6-like* at different developmental stages; (**F**) expression patterns of *Csfoxo1a-like* at different developmental stages. The letters above each chart indicate significant differences, and the bars represent standard deviations.

**Figure 5 ijms-24-07625-f005:**
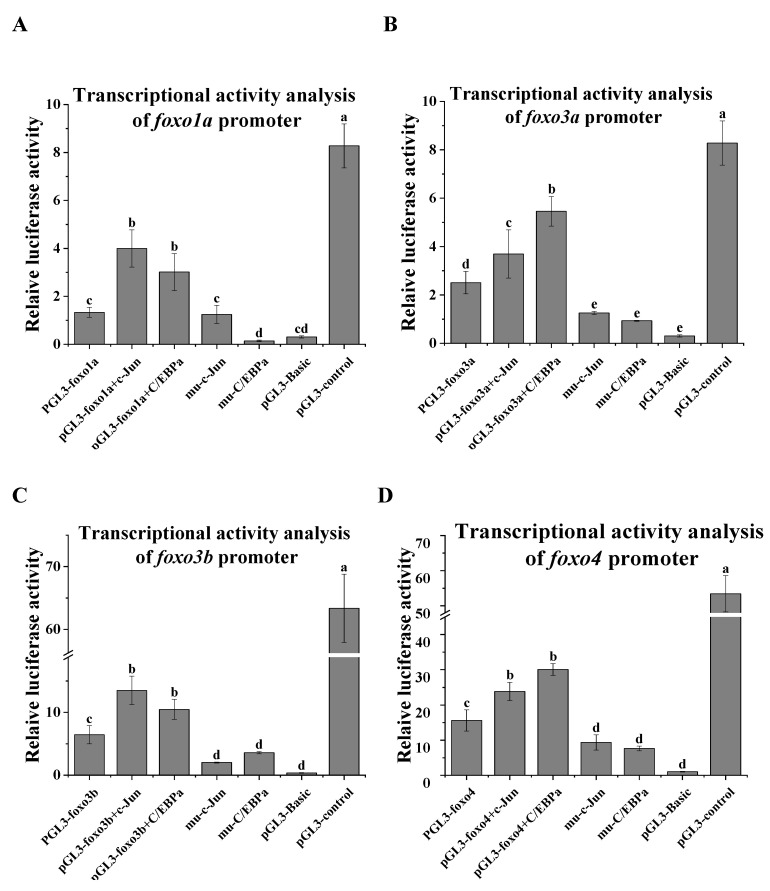
Transcriptional activity of the *Csfoxo1a*, *Csfoxo3a*, *Csfoxo3b*, and *Csfoxo4* promoters: (**A**) *Csfoxo1a*; (**B**) *Csfoxo3a*; (**C**) *Csfoxo3b*; (**D**) *Csfoxo4*. Results correspond to the addition of c-Jun and C/EBPα transcription factors, and the effect on the mutation of two transcription-factor-binding sites. Different letters indicate significant differences, and bars represent standard deviations.

**Figure 6 ijms-24-07625-f006:**
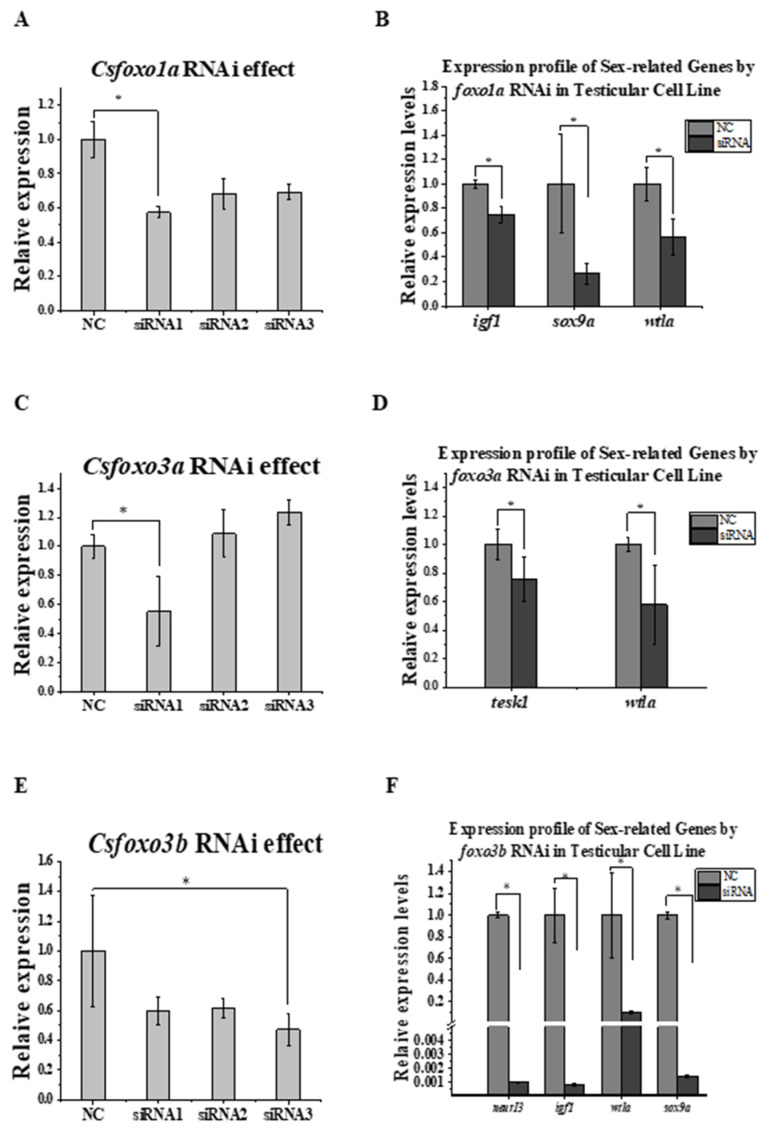
Effects of the knockdown of the genes *Csfoxo1a*, *Csfoxo3a*, and *Csfoxo3b* in a testicular cell line. (**A**) The effect of the knockdown of three siRNAs on *Csfoxo1a*. (**B**) The effect of the knockdown of *Csfoxo1a* on other genes. (**C**) The effect of the knockdown of three siRNAs on *Csfoxo3a*. (**D**) The effect of the knockdown of *Csfoxo3a* on other genes. (**E**) The effect of the knockdown of three siRNAs on *Csfoxo3b*. (**F**) The effect of the knockdown of *Csfoxo3b* on other genes. * indicates significant differences.

**Table 1 ijms-24-07625-t001:** Abbreviations used in this table: Chr, chromosome; ORF, open reading frame; AA, amino acid; MW, protein molecular weight; pI, isoelectric point.

Name	Gene ID	Chr	Genomic Location	ORF	AA	MW (kDa)	pI
*Csfoxo1a*	103378152	4	12505667-12537512	1992	663	71.49	6.29
*Csfoxo3a*	103379355	1	11149558-11159268	1872	623	66.52	5.02
*Csfoxo3b*	103380652	7	1249234-1287747	1989	662	70.12	4.79
*Csfoxo4*	103390850	15	8667641-8676081	1968	655	69.55	5.71
*Csfoxo6-like*	103387887	13	1015113-1050104	2178	725	78.07	6.78
*Csfoxo1a-like*	103395707	19	17163967-17178152	1872	623	67.9	6.6

## Data Availability

Data is contained within the article or Appendix A.

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
