# Peer review of "Identification and Functional Analysis of foxo Genes in Chinese Tongue Sole (Cynoglossus semilaevis)"

_ijms, 2023, doi:10.3390/ijms24087625_

Round 1
Reviewer 1 Report
The authors Tingting Zhang et al. have submitted a manuscript in International Journal of Molecular Sciences (Manuscript No. ijms-2287327), entitled “Identification and functional analysis of foxo genes in Chinese tongue sole (Cynoglossus semilaevis)”.
The manuscript reports a study conducted on Cynoglossus semilaevis, a traditional fish in China, with the aim of investigating foxo genes expression during development and spermatogenesis (these proteins are involved in spermatogenesis and other tissues acting as transcription factor conserved during evolution).
The authors reported the cDNA and protein of six foxo genes in Chinese tongue, all these protein share a conserved FH binding domains, which could bind the DNA target. Structure analysis and phylogenetic tree indicate that Foxo genes are conserved through vertebrate.
Their expression patterns in gonads at different developmental stages were analyzed. In addition, some genes were analized by promoter activity analysis and the siRNA-mediated knockdown in testicular cells was carried out, suggesting the regulation of foxo members in male differentiation, testis development, and spermatogenesis suggesting an important role in a wide range of organisms. These results help us to understand the functions of foxo proteins spermatogenesis in fishs. It is an interesing paper for the field.
Minor revision:
In Figure 6 the panel F is not complete, is not clear; It should be changed.
The paper is suitable for publication in International Journal of Molecular Sciences following this minor revision

Author Response
Response: Thank you for the careful review and suggestions. In Figure 6F, the values of gene expression after RNAi are very low, so it is unclear in the panel. We have changed 6F with truncated graph for improvement.
Reviewer 2 Report
fox represents the super gene family that function in many biological processes, such as foxo, foxp, foxk, foxl and so on. In sex determination and differentiation process, foxl2 and foxl3 are reported to be a key molecule. There are some reports about functionality of foxo in gametogenesis, while systematic study and comparison of all foxo members in one species are few. In the manuscript “Identification and functional analysis of foxo genes in Chinese tongue sole (Cynoglossus semilaevis)”, Zhang and colleagues have reported six foxo members and analyzed their expression pattern and promoter activity. Upon siRNA-mediated interference, the functional study has been conducted. This is a meaningful and interesting work.
The introduction has summarized the most related literatures in this field, raise unsettled questions. The materials and methods are clear and thorough enough to allow experiments to be repeated. The results are clearly and accurately illustrated. The discussion has properly interpreted the findings based on the results. In general, I think the manuscript is well written and suitable for publication, but several points need to be addressed. My specific comments are listed below.
1. In Introduction
(1) I suggest to introduce other fox gene in sex differentiation, such as foxl2 and foxl3. For six foxo members, the introduction is better to cover all member. It would be more convincing to clarify the meaning of the study.
2. In Materials and Methods,
(1) The detailed information of MS222 usage needs clarification. In addition, ethic approval information should be provided.
(2) Line 76-79, hyperlink should be removed.
3. In Results section
(1) It is better to give the similarity ratio among six members, either on nucleotide or protein level.
(2) The description of phylogenetic tree can be into more detail, not just clarify that they are grouped into four different clusters.
(3) Line 184-185, the sentence “eg. foxo6-like at 40 dpf, foxo3b and foxo4 at 60 dpf, foxo1a and foxo1a-like at 90 dpf, foxo3a at 60 dpf” needs to be re-written.
(3) The results regarding promoter activity needs to be clarified in more detail, as Figure 5 includes a lot of information.
(4) In Figure 6, please confirm whether the gene name should be neurl3.
4. In discussion,
Line 253, ifg should be igf.
5. Some grammatical points:
(1) There are some “double space”, please check and correct them.
(2) The gene or species name should appear at full name at first time, and as abbreviation afterwards.
Author Response
fox represents the super gene family that function in many biological processes, such as foxo, foxp, foxk, foxl and so on. In sex determination and differentiation process, foxl2 and foxl3 are reported to be a key molecule. There are some reports about functionality of foxo in gametogenesis, while systematic study and comparison of all foxo members in one species are few. In the manuscript “Identification and functional analysis of foxo genes in Chinese tongue sole (Cynoglossus semilaevis)”, Zhang and colleagues have reported six foxo members and analyzed their expression pattern and promoter activity. Upon siRNA-mediated interference, the functional study has been conducted. This is a meaningful and interesting work.
The introduction has summarized the most related literatures in this field, raise unsettled questions. The materials and methods are clear and thorough enough to allow experiments to be repeated. The results are clearly and accurately illustrated. The discussion has properly interpreted the findings based on the results. In general, I think the manuscript is well written and suitable for publication, but several points need to be addressed. My specific comments are listed below.
- In Introduction
(1) I suggest to introduce other fox gene in sex differentiation, such as foxl2 and foxl3. For six foxo members, the introduction is better to cover all member. It would be more convincing to clarify the meaning of the study.
Response: Thank you for the effort and suggestions.
We have added the information of foxl2 and foxl3, please refer to Line 64-65.
The introduction of foxo4 and foxo6 has been added, please refer to Line 59-61
We have described the meaning for conducting this study, please refer to Line 62-63.
- In Materials and Methods,
(1) The detailed information of MS222 usage needs clarification. In addition, ethic approval information should be provided.
Response: Thank you. We have added the related information in Line 226-227 and 233-234.
(2) Line 76-79, hyperlink should be removed.
Response: Thank you. We have removed the hyperlink, refer to Line 214-216.
- In Results section
(1) It is better to give the similarity ratio among six members, either on nucleotide or protein level.
Response: We have made the multiple alignment, and the similarity of six Foxo proteins are 53.1%. The information was added in the results (Line 91-92).
(2) The description of phylogenetic tree can be into more detail, not just clarify that they are grouped into four different clusters.
Response: We have expanded the description in Line 184-186.
(3) Line 184-185, the sentence “eg. foxo6-like at 40 dpf, foxo3b and foxo4 at 60 dpf, foxo1a and foxo1a-like at 90 dpf, foxo3a at 60 dpf” needs to be re-written.
Response: It has been revised into “e.g. foxo6-like at 40 dpf, foxo3a, fox3b and foxo4 at 60 dpf, foxo1a and foxo1a-like at 90 dpf”
(3) The results regarding promoter activity needs to be clarified in more detail, as Figure 5 includes a lot of information.
Response: Thanks, we have clarified the Figure 5 into more detail, please see Line 136-137.
(4) In Figure 6, please confirm whether the gene name should be neurl3.
Response: Thank you for pointing out the mistake, it should be neurl3 and we have made the correction.
- In discussion,
Line 253, ifg should be igf.
Response: Thank you. We have corrected the mistake.
- Some grammatical points:
(1) There are some “double space”, please check and correct them.
(2) The gene or species name should appear at full name at first time, and as abbreviation afterwards.
Response: Thank you. For above-mention two issues, we have carefully checked the manuscript and sent it for linguistic improvement (Certification provided as attachment).
Reviewer 3 Report
The author identified 6 members of foxo gene in Tongue Sole and identified their transcription enhancer factor (C/EBPα, C-Jun), expression stage, downstream regulatory genes, which suggest that foxo has important functions in male differentiation.
1. In real-time PCR assay, what kind of reference gene was used for calculation?
2. I am not sure whether it is meaningful to perform promoter activity analysis. The activity of promoter is largely affected by the sequence cloned and cell lines.
3. What kind of testicular cell line was used in this study? Whether this cell line includes Sertoli cell, Leydig cell or different stage sperms?
4. I noticed that RNAi efficacy is not so good. The authors employed three different siRNAs, but only one is effective. Why?
5. For evaluating the effect of RNAi on testicular cell line, the authors only select some target gene randomly. Is it possible to use RNA sequencing to see the global effect of RNAi?
6. I noticed the significant effect of foxo3b RNAi on the expression of nerul3, igf1, wt1a and sox9a. I am worried that this effect is due to the toxicity of this siRNA.
7. The author mentioned that all motifs except motif 9 can be detected in all foxo members, but in Figure 3, motif 10 was not found in foxo4.
8. The author employed phylogenetic tree analysis and structure analysis to prove that foxo are highly conserved in vertebrate. The phylogenetic tree analysis includes mammals, but the structural analysis only includes bony fish. Why?
9. I noticed that the author did not use all the members of foxo for promoter activity analysis and knockdown assay, only selected a few of them. It is recommended to apply these analyses to all foxos. Otherwise, the author should provide a reason.
10. Is it possible to use in situ hybridization to see the expression pattern of foxos gene in testis?
11. Many spelling errors and grammar problems could be found.
Author Response
The author identified 6 members of foxo gene in Tongue Sole and identified their transcription enhancer factor (C/EBPα, C-Jun), expression stage, downstream regulatory genes, which suggest that foxo has important functions in male differentiation.
- In real-time PCR assay, what kind of reference gene was used for calculation?
Response: We are sorry for the negligence. β-actin was used as internal reference, the information was added in Line 247.
- I am not sure whether it is meaningful to perform promoter activity analysis. The activity of promoter is largely affected by the sequence cloned and cell lines.
Response: Thank you for the professional advice. The promoter activity is closely related to gene transcription and the activity analysis is widely used. In our study, we have also included transcription factors, Stat5a and Pou1f1a, which showed no effect on foxo promoter, so the results are rather reliable. We totally agree that the analysis is largely affected by the sequence cloned and cell lines, and we would perform in-depth study such as in vivo promoter activity analysis in future.
- What kind of testicular cell line was used in this study? Whether this cell line includes Sertoli cell, Leydig cell or different stage sperms?
Response: The predominant type is Sertoli cells as germ cells can hardly maintain after passing many generations. We have clarified this information in Materials and Methods (Line 280-281).
- I noticed that RNAi efficacy is not so good. The authors employed three different siRNAs, but only one is effective. Why?
Response: The RNAi effect could differ for different. For each gene, three siRNA targeting different sites were designed and the siRNA with the best knockdown effect was selected for further study. We have also added the information in Line 278-279.
- For evaluating the effect of RNAi on testicular cell line, the authors only select some target gene randomly. Is it possible to use RNA sequencing to see the global effect of RNAi?
Response: Thank you for the advice. As above mentioned, the testicular cell line mainly consist of Sertoli cells and the number of transcribed genes are preliminary, so these targets are selected according to our previous experience.
- I noticed the significant effect of foxo3b RNAi on the expression of nerul3, igf1, wt1a and sox9a. I am worried that this effect is due to the toxicity of this siRNA.
Response: It is a wonderful question. In fact, we could not totally exclude this possibility, but it could be subtle even there is toxicity because of two reasons. First, we have also examined the transcription level of other genes such as star and p450scc, while they showed no significant change. Second, the growth of cell were normal after transduction.
- The author mentioned that all motifs except motif 9 can be detected in all foxo members, but in Figure 3, motif 10 was not found in foxo4.
Response: It is our mistake. We have re-written the description from Line 105-111. “As shown in Figure 3, a total of 12 motifs were detected, and the signature sequences of the 12 motifs are shown in Supplementary Table 2. Motifs 1 and 2 are high-ly conserved FH domains, and motif 9 is only found in Foxo3 and Foxo4 of bony fish and consists of 50 amino acids. In fish, Foxo3 contains all motifs, while Foxo4 lacks motif 10, and Foxo1 and Foxo6 lack motif 9. In mammals, the Foxo motifs differ among humans, mice, and rats. In general, mammalian Foxo1 and Foxo3 lack motif 11, Foxo4 lacks motif 8, and Foxo6 lacks motifs 8, 10, and 12”.
- The author employed phylogenetic tree analysis and structure analysis to prove that foxo are highly conserved in vertebrate. The phylogenetic tree analysis includes mammals, but the structural analysis only includes bony fish. Why?
Response: Thank you very much. In fact, the conservation of Foxo are mainly based on sequence alignment and phylogenetic tree analysis. The structure or motif analysis are conducted for detailed illustration. According to your nice advice, we have added structural analysis for mammals as Supplementary Figure 3 and result section (Line 109-111).
- I noticed that the author did not use all the members of foxo for promoter activity analysis and knockdown assay, only selected a few of them. It is recommended to apply these analyses to all foxos. Otherwise, the author should provide a reason.
Response: Thank you. We have performed the promoter activity and knockdown experiments, while there are no significant effect, so the data were not shown.
- Is it possible to use in situ hybridization to see the expression pattern of foxos gene in testis?
Response: Thanks a lot. It is a very good point. Actually, we have tried in situ hybridization for fox1a, 3a and 3b in testis. However, due to the similarity of the six members, the probe design was not easy (there is no signal when the probes are short and specific, while sequences overlap among different members when probes are long) and the results are not satisfactory. We are considering to design probes at UTR regions and is still under the way.
- Many spelling errors and grammar problems could be found.
Response: We have checked the manuscript and sent for the linguistic improvement, and hope to reach the journal requirement.